# Somatic Embryogenesis Initiation in Sugi (Japanese Cedar, *Cryptomeria japonica* D. Don): Responses from Male-Fertile, Male-Sterile, and Polycross-Pollinated-Derived Seed Explants

**DOI:** 10.3390/plants10020398

**Published:** 2021-02-19

**Authors:** Tsuyoshi E. Maruyama, Saneyoshi Ueno, Yoshihisa Hosoi, Shin-Ichi Miyazawa, Hideki Mori, Takumi Kaneeda, Yukiko Bamba, Yukiko Itoh, Satoko Hirayama, Kiyohisa Kawakami, Yoshinari Moriguchi

**Affiliations:** 1Department of Research Planning and Coordination, Forestry and Forest Products Research Institute, Matsunosato 1, Tsukuba 305-8687, Japan; 2Department of Forest Molecular Genetics and Biotechnology, Forestry and Forest Products Research Institute, Matsunosato 1, Tsukuba 305-8687, Japan; saueno@ffpri.affrc.go.jp (S.U.); yh2884@ffpri.affrc.go.jp (Y.H.); miyashin@ffpri.affrc.go.jp (S.-I.M.); morih@ffpri.affrc.go.jp (H.M.); 3Graduate School of Science and Technology, Niigata University, Ikarashi 8050, Niigata 950-2181, Japan; takumi.kane01@gmail.com (T.K.); chimori@agr.niigata-u.ac.jp (Y.M.); 4Niigata Prefectural Forest Research Institute, Unotoro 2249-5, Niigata 958-0264, Japan; bamba.yukiko@pref.niigata.lg.jp (Y.B.); itou.yukiko@pref.niigata.lg.jp (Y.I.); 5Agriculture and Forestry Promotion Department, Niigata Regional Promotion Bureau, Niigata Prefectural Government, Hodojima 2009, Niigata 956-8635, Japan; cosato310@gmail.com; 6Verde Co., Ltd., Kitagaya 244, Aichi 441-8123, Japan; k.kawakami@verde-agribio.co.jp

**Keywords:** Cupressaceae, embryogenic cell induction, EM medium, megagametophyte, pollen-free, tissue culture

## Abstract

This study aimed to obtain information from several embryogenic cell (EC) genotypes analyzing the factors that affect somatic embryogenesis (SE) initiation in sugi (*Cryptomeria japonica,* Cupressaceae) to apply them in the improvement of protocols for efficient induction of embryogenic cell lines (ECLs). The results of several years of experiments including studies on the influence of initial explant, seed collection time, and explant genotype as the main factors affecting SE initiation from male-fertile, male-sterile, and polycross-pollinated-derived seeds are described. Initiation frequencies depending on the plant genotype varied from 1.35 to 57.06%. The best induction efficiency was achieved when seeds were collected on mid-July using the entire megagametophyte as initial explants. The extrusion of ECs started approximately after 2 weeks of culture, and the establishment of ECLs was observed mostly 4 weeks after extrusion on media with or without plant growth regulators (PGRs). Subsequently, induced ECLs were maintained and proliferated on media with PGRs by 2–3-week-interval subculture routines. Although, the initial explant, collection time, and culture condition played important roles in ECL induction, the genotype of the plant material of sugi was the most influential factor in SE initiation.

## 1. Introduction

Somatic embryogenesis (SE) is currently the most efficient technique for large-scale propagation in clonal forestry and a powerful tool for plant regeneration in basic biological studies, genetic engineering, and implementation of multi-varietal forestry [1,2,3]. However, a low SE initiation rate is one of the major problems in using SE in some species and among individuals and families within the same species [4,5]. Initial explant selection has been a critical factor for SE initiation in conifers [6]. In general, precotyledonary and cotyledonary zygotic embryos are reported as the best initial explants for pine and spruce species, respectively [7]. These results indicate that the timing of the explant excision when zygotic embryos are highly responsive to inducing embryogenic cells (ECs) is the key to a successful SE initiation [8]. However, for many species, the time for a successful SE initiation is limited to a very short period of only a few weeks each year [9]. In addition, optimal seed collection time as a critical factor in SE initiation (particularly in open-pollinated cones) may also be influenced by weather, location, and cross variation [10,11,12,13]. Regardless of method, enhancing the success rate in the initiation stage is of vital importance for the improvement of SE protocols in breeding programs.

Sugi is the most important forestry species in Japan, representing 44% of the total reforested area. However, more than 30 million people (>30% of the total population) suffer from sugi pollinosis, a serious social and public health problem. As a countermeasure against sugi pollinosis, the use of male-sterile plants (MSPs; pollen-free plants) is a recommended alternative. Currently, MSPs of sugi are produced mostly by artificial crossing between a male-sterile tree (*ms1/ms1*) and an elite tree (a superior tree selected after growth performance and morphological traits) heterozygous for *MS1* (*Ms1*/*ms1*) [14]. The *MS1* is the most representative male sterility gene found so far in pollen-free individuals of sugi [15]. The selection of MSPs is carried out after inducing male flowering by applying gibberellin, a plant growth regulator that induces flowering in sugi [16,17]. Using this method, approximately ≥50% of seedlings do not become male-sterile due to the disregarded law of segregation, making production efficiency extremely poor [18]. In this context, the development of technology to produce excellent MSPs in a short period of time is one of the priority goals. At present, efficient protocols to propagate male-sterile somatic plants of sugi combine a selection of ECs with marker-assisted selection and propagation via SE has been established [19]. On the other hand, as one of the methods for labor-saving production of genetically diverse seedlings, it is conceivable to use seeds derived from polycross pollination, artificial mating using mixed pollen from multiple parents. Although the pollen parent contribution rate of mature seeds derived from polycross pollination has been reported to be significantly different from the expected value [20], there is no case study of pollen parent contribution rate in each process of SE using immature seeds of sugi derived from polycross pollination.

Here, we report the results of several years of experiments on SE initiation in sugi, including studies on the influence of initial explant, seed collection time, and explant genotype, as the main factors affecting SE initiation from male-fertile, male-sterile, and polycross-pollinated-derived seeds. This study aimed to obtain information from several seed families or crosses and to analyze the factors that affect the initial stage of SE process in sugi.

## 2. Results and Discussion

### 2.1. Somatic Embryogenesis Initiation from Male-Fertile-Derived Seed Explants

#### 2.1.1. Effect of Culture Media and Seed Collection Times

The first experimental approach was to determine the effect of plant growth regulators (PGRs) and seed collection times on SE initiation. The entire megagametophyte containing the zygotic embryo at different developmental stages was used as an initial explant. The extrusion of ECs (Figure 1A) started approximately after 2 weeks of culture, and the establishment of embryogenic cell lines (ECLs) was observed approximately 4 weeks after extrusion (Figure 1B). Initiation frequencies varied according to culture medium and seed collection time from 1.33 to 14.67% (Table 1). The lowest average frequency for collected explants was achieved in mid-June (2.67%), increasing to 6.67% and 10.67% for seed collection in late-June and early-July, respectively, and was highest (13.33%) in mid-July. The statistical analysis indicated that the proportion of explants with SE initiation response significantly differed among seed collection times (χ^2^ = 12.829, df = 3, *p* < 0.05 for mid-June, and *p* < 0.1 for late-July), but no significant differences were observed between the late-June and early-July collection times. Although the best response for both culture media was recorded in mid-July (Table 1), germination and SE initiation were observed simultaneously in a number of explants cultured on medium without PGRs (Figure 1C). This result shows that seeds collected in mid-July can germinate in vitro, suggesting that they reach physiological maturity. Successful SE initiation on media without PGRs has been reported also in pines [21,22,23,24] and some cypress species [25,26,27]. These results confirm that PGR supplementation to the medium is not necessary when the explants are in an appropriate developmental stage. Nevertheless, even though for sugi seed explants collected from mid-June to mid-July the exogenous PGRs in the culture medium were not a critical factor for the induction of ECs, mid-July can be the critical collection time limit for SE initiation on media with no PGRs. Therefore, although the statistical analysis result indicated that the SE initiation frequencies between media with or without PGRs were no significantly different (χ^2^ = 0.79339, df = 1, *p* > 0. 05), for further SE initiation experiments, media supplemented with 10 μM 2,4-dichlorophenoxyacetic acid (2,4-D) and 5 μM 6-benzylaminopurine (BA) were used.

Even though different initial culture media and seed sources were used, the highest SE initiation frequencies in sugi (20–35%) with seeds collected in mid-July was also reported by Ogita et al. [28]. Similarly, Taniguchi and Kondo [29] also reported that seeds collected in mid-July were the best explants for SE initiation in sugi, recording induction rates of up to 33.3%, testing 20 different families from the open-pollinated (OP) seed orchard.

#### 2.1.2. Effect of Initial Explants

As shown in Table 2, although SE initiation from whole seed (Figure 2A) and from seeds with coat cut lengthwise (Figure 2B) was observed, the mean initiation frequencies from all mother trees were very low, reaching only 4.17% and 5.00%, respectively, compared to the best induction rate (49.19%) achieved using megagametophyte explants. The statistical analysis indicated that the proportion of explants with SE initiation response significantly differed among explant types (χ^2^ = 141.92, df = 2, *p* < 0. 001) and seed families (χ^2^ = 24.414, df = 4, *p* < 0. 05) except for the “Taga 4” and “Tsukuba 2” families. As widely known, the choice of appropriate explant is a determining factor for successful SE initiation. In this way, the entire megagametophyte preferably from immature seeds was the most common explants reported in conifers [30,31,32,33,34]. Similarly, the highest induction frequencies for sugi were achieved with entire megagametophytes as initial explants (Table 2). This method has the advantage that it avoids cumbersome extraction of tiny immature embryos. The use of whole seed as initial explants for SE initiation are not commonly reported and possibly limited to species with very small seeds that are difficult to handle [35]. Although the use of whole seeds offers the potential for less handling and time-consuming in preparing initial explants, the result of our experiment suggests that this method is not appropriate for sugi.

### 2.2. Somatic Embryogenesis Initiation from Male-Sterile-Derived Seed Explants

#### 2.2.1. Effect of Seed Collection Times

SE initiation responses from sugi seed families carrying the male sterility gene *MS1* at different seed collection times are shown in Table 3. The statistical analysis results indicated that the proportion of explants with SE initiation response significantly differed among seed families (χ^2^ = 477.1, df = 4, *p* < 0.001) and seed collection times (χ^2^ = 63.272, df = 2, *p* < 0.001) with the exception of collection in late-July (Table 3). The SE initiation rate varied from 0.62 to 59.03% with the average frequencies of 19.32%, 32.00%, and 25.26% for early-July, mid-July, and late-July, respectively. The best result recorded for the mid-July collection was concordant with the experiment results with male-fertile-derived seed explants.

#### 2.2.2. Effect of Initial Explants

Table 4 shows the SE initiation responses of different initial explants from sugi seed families carrying the male sterility gene *MS1*. Similar to the experiment results with male-fertile-derived seed explants, the highest initiation frequency (30.57%) was achieved with megagametophytes, whereas the lowest rate (0.46%) was recorded when whole seeds were used as initial explants. The results of the statistical analysis indicated that the proportion of explants with SE initiation response significantly differed among initial explant types (χ^2^ = 890.77, df = 4, *p* < 0.001) and seed families (χ^2^ = 693.45, df = 3, *p* < 0.001). The induction of ECs from seeds stored at 5 °C for 1 and 4 weeks was achieved but with no improvement in SE initiation frequencies was observed compared to non-stored megagametophyte explants (Table 4). In contrast, cold storage of initial explants improved initiation frequencies in white spruce [36] and radiata pine [37]. The difference in these results could be attributed to the cold storage method used since, while the seeds of white spruce and radiata pine were stored before dissection, the seeds of sugi were stored after surface sterilization in our experiment. Similarly, Haggman et al. [38] reported that, although the cold treatment of the cones had no significant effect on SE initiation in Scots pine, the cones can be collected and stored for at least 2 months without losing the ability to initiate SE. Park [39] also reported that eastern white pine cones might be stored at 3 °C for at least 40 days without reducing embryogenic capacity. These results suggest that cold storage of explants could be a promising alternative for practical uses to improve induction rates and to extend the very narrow window of time when SE initiation is possible. Although in our experiments the cold storage treatment of the explant did not increase the SE initiation frequencies, more research is needed to clarify the potential of cold preconditioning of initial explants in sugi. Improving the cold preconditioning techniques of initial explants to increase induction frequencies (to capture as many genotypes as possible) and to extend SE initiation period is important to develop varietal lines and to manage genetic diversity [11].

On the other hand, in contrast to our expectation, seeds with the coat cut lengthwise as initial explants showed low induction frequencies (4.73–5.00%) comparable with whole seed explants (0.46–4.17%). This result can be attributed to the improper cutting technique applied, which possibly did not allow the extrusion of ECs. More effort is needed to improve this technique to save energy and time in preparing the initial material.

### 2.3. Somatic Embryogenesis Initiation from Polycross-Pollinated-Derived Seed Explants

#### Effect of Polycross Family and Seed Collection Times

Seeds derived from polycross pollination (artificial mating using mixed pollen of three and 10 parents) were used as the initial explant for SE initiation in sugi. As shown in Table 5, although initiation frequencies varied according to the polycross family and seed collection time, high induction rates ranging from 38.86–71.18% (with an overall average of 54.50%) were achieved. The results of the statistical analysis indicated that the proportion of explants with SE initiation response significantly differed between the polycross family (χ^2^ = 28.997, df = 1, *p* < 0.001) and among seed collection times (χ^2^ = 178.63, df = 2, *p* < 0.001). Similar to the results obtained in the experiments with male-fertile and male-sterile-derived seed explants, the best responses from polycross families were recorded with plant materials collected in mid-July (Table 5). These results confirmed that, during mid-July, the zygotic embryos are highly responsive to induce ECs. On the other hand, the results regarding the effect of the polycross family indicated that the SE initiation frequency of seeds derived from mixed pollen of three parents was better than 10 parents. To clarify this response, studies are currently being carried out with molecular markers. Preliminary results indicate that the highest induction frequency achieved with three mix-pollinated-derived seeds is subordinate to the dominance of a specific parent. In contrast, the dominance of specific pollen parents was suppressed by using 10 mix-pollinated-derived seeds [40]. Even though more studies are needed to better understand the mechanism of polycross fertilization and its effect on SE process, these results suggest that polycross pollination using many individuals could be a practical alternative for the production of seedlings with high genetic diversity.

### 2.4. Summarized Results on SE Initiation in Sugi from Male-Fertile, Male-Sterile, and Polycross-Pollinated-Derived Seeds

The results of several years of experiments on SE initiation in sugi from male-fertile, male-sterile, and polycross-pollinated-derived seeds are summarized in Table 6. These initiation frequencies were consistent with the reports on SE in artificially pollinated seed families of sugi [41]. The best average induction frequency was achieved with polycross-pollinated-derived seeds (45.20%). However, despite the results suggesting that the statistical analysis indicated that the proportion of explants with SE initiation response significantly differed among the origen of families (χ^2^ = 618.55, *df* = 2, *p* < 0.001), it is important to note that the best average rate obtained with polycross-pollinated-derived seeds were the results of only two years of collection. In addition, SE initiation frequencies from male-fertile, male-sterile, and polycross-pollinated-derived seeds showed great variation ranging from 1.35–67.46% (Appendix A), 2.78–40.90% (Appendix A), and 8.37–58.67% (Appendix A), respectively. Therefore, in our opinion, these differences are attributable to the genotypic ability to induce a large number of ECs achieved in “S 11” × “3 Mix” (57.06%) and “S 11” × “10 Mix” (49.85%) families (Table 6). This result suggests that the observed differences in the efficiency of SE initiation among male-fertile, male-sterile, and polycross-derived families can be attributed to the genotype of the plant material regardless of its origin. The genotype of initial explant has been reported as the most influential factor in SE initiation in a number of conifer species [42,43,44,45].

### 2.5. Maintenance and Proliferation of Embryogenic Cells

Cultures for the maintenance and proliferation of ECs were carried out at 2–3 week intervals. Despite differences among ECLs with regard to the proliferation rate and morphological structure observed (Figure 3), the culture medium was able to support the growth of almost all lines by subculture routines for several years without losing their proliferation potential and initial morphological characteristics, as described elsewhere [10]. Some lines have been maintained and proliferated for more than 10 years, although with differences among genotypes regarding plant conversion capacity [46]. The embryogenesis response of sugi largely differs among ECLs [47].

## 3. Materials and Methods

### 3.1. Plant Material

Sugi seeds collected from seed orchards were used as plant material for the SE initiation experiments. At each collection time, the samples of zygotic embryos were observed to determine their developmental stage according to the scale used to classify zygotic embryo development in loblolly pine [48]. The developmental stage of explants collected from mid-June to early-July was the pre-embryo stage equivalent to stages 1–3. Collections in mid-July were mostly represented by early embryo stages equivalent to stages 3–5, and seeds collected in late-July showed the pre-cotyledonary stages equivalent to stages 6–8. Except experiments with different explant types, in all the other experiments, the entire megagametophyte was used as the initial explant.

#### 3.1.1. Somatic Embryogenesis Initiation from Male-Fertile-Derived Seed Explants

Male-fertile-derived seeds were collected from seventeen different OP mother trees in seed orchards at the Forestry and Forest Products Research Institute (Tsukuba, Ibaraki, Japan) and Ibaraki Prefectural Government Forestry Technology Center (Naka, Ibaraki, Japan) from 1997 to 2011 (Appendix A). To determine the effect of seed collection time and culture media on SE initiation, four seed collections from OP Yanase 104 mother tree were carried out at approximately two-week intervals from mid-June to mid-July (1997) and cultured on medium with or without PGRs (Table 1). The effect of different initial explants on SE initiation was tested with collected plant materials in early-July (2005) from six different OP mother trees (Table 2).

#### 3.1.2. Somatic Embryogenesis Initiation from Male-Sterile-Derived Seed Explants

Male-sterile-derived seeds were collected from eight different full-sib seed families carrying the male sterility gene *MS1 or MS2* [49] in seed orchards at Niigata Prefectural Forest Research Institute (Murakami, Niigata, Japan) from 2016 to 2018 (Appendix A). The effect of collection time on SE initiation was evaluated with seeds from four different seed families collected from early to late-July in 2016 and 2017 (Table 3). To determine the effect of initial explants on SE initiation, five different initial explants (including megagametophyte, megagametophyte isolated from seeds stored at 5 °C for 1 to 4 weeks, whole seeds, and seeds with coat cut lengthwise) collected in 2017 from four different seed families carrying the male sterility gene *MS1* were tested (Table 4).

#### 3.1.3. Somatic Embryogenesis Initiation from Polycross-Pollinated-Derived Seed Explants

Polycross-pollinated-derived seeds were collected from five different full-sib seed families using three parents of mixed pollen (3 mix) and nine or ten parents of mixed pollen (9 mix or 10 mix) in a seed orchard at the Niigata Prefectural Forest Research Institute (Murakami, Niigata, Japan) from 2019 to 2020 (Appendix A). The effect of collection time on SE initiation was evaluated using 3 mix and 10 mix polycross-pollinated-derived seeds collected from early to late-July in 2019 (Table 5).

### 3.2. Surface Sterilization of Seeds

After isolation from collected cones, the seeds were surface sterilized with 1% (*w/v* available chlorine) sodium hypochlorite solution for 15 min and then rinsed three times with sterile distilled water for 5 min each time.

### 3.3. Media and Culture Conditions

For the induction of ECs, explants were placed horizontally onto initiation media contained in 90 × 15 mm quad-plates (three explants per well, 12 per plate) and cultured in darkness at 25 °C. The initiation medium containing basal salts reduced to half the concentration from the standard EM medium [10] was supplemented with 10 g L^−1^ sucrose, 10 μM 2,4-D, 5 μM BA, 0.5 g L^−1^ casein acid hydrolysate, and 0.5 g L^−1^ glutamine and was solidified with 3 g L^−1^ gellan gum (Gelrite^®^; Wako Pure Chemical, Osaka, Japan). The pH was adjusted to 5.8 prior to autoclaving the medium for 15 min at 121 °C. Media without PGRs but containing 2 g L^−1^ activated charcoal (Wako Pure Chemical, Osaka, Japan) were also tested to compare SE initiation frequencies with those media containing PGRs (Table 1). For all other experiments, media supplemented with 2,4-D and BA were used.

### 3.4. Maintenance and Proliferation of Embryogenic Cells

Induced ECs were subcultured every 2–3 weeks on maintenance/proliferation medium containing basal salts reduced to half the concentration from the standard EM medium [10] supplemented with 3 μM 2,4-D, 1 μM BA, 30 g L^−1^ sucrose, 1.5 g L^−1^ glutamine, and 3 g L^−1^ gellan gum. Clumps of embryogenic cells (12 per plate) were cultured in darkness at 25 °C.

### 3.5. Statistical Analysis

The proportion differentiation of the explants with SE initiation response among seed families, seed collection times, and initial explant types were examined using Pearson’s Chi-squared test. To further elucidate which part of the data was causing the significant differentiation, the residuals of the Chi-squared test were used to conduct the post hoc analysis and the *p*-values were adjusted with a Bonferroni correction [50]. Pearson’s Chi-squared test was performed using R version 3.6.2 [51], and the post hoc analysis based on the residuals of the Chi-squared test was performed using the R package “chisq.posthoc.test” [52].

## 4. Conclusions

Our research in sugi proved that, although SE initiation was possible from mid-June to late-July, the best induction efficiency was achieved when seeds were collected in mid-July. The best collection time for SE initiation was confirmed in experiments with male-fertile, male-sterile, and polycross-pollinated-derived seed explants. Notwithstanding differences regarding SE initiation frequencies among families observed throughout our experiments, the optimal collection time for almost all seed families was determined around mid-July. Similarly, as reported for other conifers, the megagametophyte explant was also the best plant material for SE initiation in sugi. However, even though the initial explant, collection time, and culture condition played important role in ECL induction, the genotype of the plant material of sugi was the most influential factor in SE initiation. More effort is necessary to obtain experimental information about the SE initiation performance of sugi genotypes using control-pollinated families to select the most appropriate female and male parents. Emphasizing this point, we believe that the polycross pollination technique can be a practical tool for this purpose.

## Figures and Tables

**Figure 1 plants-10-00398-f001:**
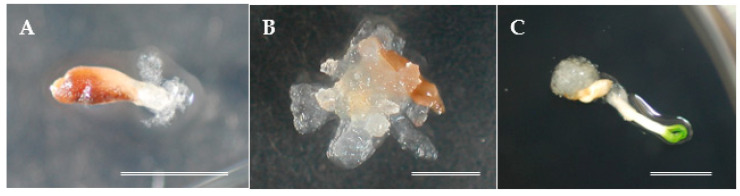
Somatic embryogenesis (SE) initiation in sugi: (**A**) extrusion of embryogenic cells (ECs) about 2 weeks after culture initiation on media with plant growth regulators (PGRs), (**B**) EC proliferation on a medium with PGRs approximately 4 weeks after extrusion, and (**C**) germination and SE initiation on a medium with no PGRs approximately 4 weeks after culture initiation. Bars: 5 mm.

**Figure 2 plants-10-00398-f002:**
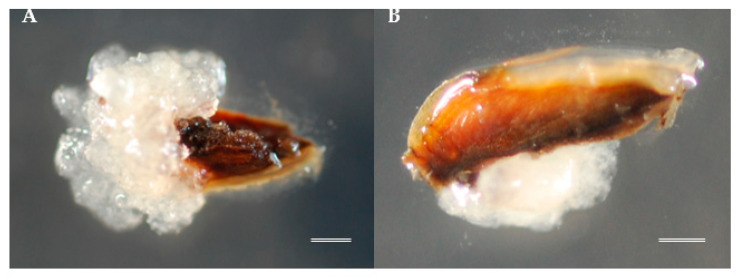
Somatic embryogenesis (SE) initiation in sugi: (**A**) embryogenic cell (EC) proliferation from whole seed explant on media with plant growth regulators (PGRs) approximately 6 weeks after culture initiation and (**B**) from seeds with coat cut lengthwise on media with PGRs approximately 4 weeks after culture initiation. Bars: 1 mm.

**Figure 3 plants-10-00398-f003:**
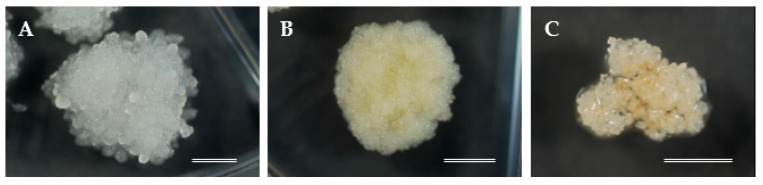
Proliferation of embryogenic cell lines (ECLs) with different morphological structures: (**A**) mucilaginous whitish embryogenic cells (ECs), (**B**) mucilaginous yellowish ECs, and (**C**) friable white translucent ECs. Bars: 5 mm.

**Table 1 plants-10-00398-t001:** Somatic embryogenesis (SE) initiation frequency from open-pollinated-derived seeds of the male-fertile family of sugi. The data represent the explants with SE initiation response and the total number of explants tested, and the number in the parentheses represent the initiation frequency (%) for each culture medium with or without plant growth regulators (PGRs) at different seed collection times.

Mother Tree	Culture Medium	SE initiation Frequency (%) by Seed Collection Time				
		Mid-June	Late-June	Early-July	Mid-July	Total
“Yanase 104”	PGRs (−)	1/75 (1.33)	3/60 (5.00)	6/60 (10.00)	9/75 (12.00)	19/270 (7.04) ns
“Yanase 104”	PGRs (+)	3/75 (4.00)	7/90 (7.78)	10/90 (11.11)	11/75 (14.67)	31/330 (9.39) ns
Total		4/150 (2.67) *	10/150 (6.67) ns	16/150 (10.67) ns	20/150 (13.33) ^☥^	50/600 (8.33)

PGRs (−): explants cultured on medium without plant growth regulators; PGRs (+): explants cultured on medium with plant growth regulators; ns: no significant differentiation at *p* > 0.05 by post hoc analysis of Pearson’s Chi-squared test; * significantly different at *p* < 0.05 by post hoc analysis of Pearson’s Chi-squared test; ☥ significantly different at *p* < 0.1 by post hoc analysis of Pearson’s Chi-squared test.

**Table 2 plants-10-00398-t002:** Somatic embryogenesis (SE) initiation frequency from open-pollinated-derived seeds of male-fertile families of sugi. The data represent the explants with SE initiation response and the total number of explants tested, and the initiation frequency (%) for six mother trees using different initial explants.

Mother Trees	SE Initiation Frequency (%) by Initial Explant Type			
	Megagametophyte	Whole Seed	Seed with Coat Cut Lengthwise	Total
“Kuji 14”	21/48 (43.75)	2/24 (8.33)	3/24 (12.50)	26/96 (27.08) *
“Naka 3”	6/48 (12,50)	0/24 (0.00)	0/24 (0.00)	6/96 (6.25) *
“Nihari 2”	22/48 (45.83)	1/24 (4.17)	3/24 (12.50)	26/96 (27.08) *
“Taga 2”	170/252 (67.46)	ND	ND	170/252 (67.46) NT
“Taga 4”	9/48 (18.75)	1/24 (4.17)	0/24 (0.00)	10/96 (10.42) ns
“Tsukuba 2”	14/48 (29.17)	1/24 (4.17)	0/24 (0.00)	15/96 (15.63) ns
Total	242/492 (49.19) ***	5/120 (4.17) ***	6/120 (5.00) ***	253/732 (34.56)

ND: no data; NT: not tested; ns: no significant differentiation at *p* > 0.05 by post hoc analysis of Pearson’s Chi-squared test; * significantly different at *p* < 0.05 by post hoc analysis of Pearson’s Chi-squared test; *** significantly different at *p* < 0.001 by post hoc analysis of Pearson’s Chi-squared test.

**Table 3 plants-10-00398-t003:** Somatic embryogenesis (SE) initiation frequency of sugi seed family carrying the male sterility gene *MS1*. The data represent the explants with SE initiation response and the total number of explants tested, and the number in the parentheses represent the initiation frequency (%) for each of four seed families at different seed collection times.

Seed Family ^$^	Collection Year	SE Initiation Frequency (%) by Seed Collection Time			
		Early-July	Mid-July	Late-July	Total
♀ “Shindai 3” ♂ “Suzu 2”	2016	108/300 (36.00)	59/420 (14.05)	45/516 (8.72)	212/1236 (17.15) ***
♀ “Shindai 3” ♂ “Suzu 2”	2017	54/156 (34.62)	191/383 (49.87)	39/168 (23.21)	284/707 (40.17) ***
♀ “Fukushima-funen 1” ♂ “S3-37(1)”	2017	2/324 (0.62)	62/516 (12.02)	17/216 (7.87)	81/1056 (7.67) ***
♀ “Fukushima-funen 1” ♂ “Oi 7”	2017	11/156 (7.05)	259/636 (40.72)	85/144 (59.03)	355/936 (37.93) ***
♀ “Fukushima-funen 1” ♂ “S3-118(2)”	2017	29/120 (24.17)	191/429 (44.52)	196/468 (41.88)	416/1017 (40.90) ***
Total		204/1056 (19.32) ***	762/2384 (32.00) ***	382/1512 (25.26) ns	1348/4952 (27.22)

**^$^**: All female and male parents had *ms1/ms1* and *Ms1/ms1* genotype, respectively; ns: no significant differentiation at *p* > 0.05 by post hoc analysis of Pearson’s Chi-squared test; *** significantly different at *p* < 0.001 by post hoc analysis of Pearson’s Chi-squared test.

**Table 4 plants-10-00398-t004:** Somatic embryogenesis (SE) initiation frequency of sugi seed families carrying the male sterility gene *MS1*. The data represent the explants with SE initiation response and the total number of explants tested, and the initiation frequency (%) for each of four seed families using different initial explants.

Seed Family ^$^	SE Initiation Frequency (%) by Initial Explant Type					
	Megagametophyte	Megagametophyte from 5 °C (1 week)	Megagametophyte from 5 °C (4 weeks)	Whole Seed	Seed with Coat Cut Lengthwise	Total
♀ “Shindai 3” ♂ “Suzu 2”	284/707 (40.17)	402/1416 (28.39)	56/435 (12.87)	0/48 (0.00)	1/48 (2.08)	743/2654 (28.00) ***
♀ “Fukushima-funen 1” ♂ “S3-37(1)”	81/1056 (7.67)	66/932 (7.08)	36/635 (5.67)	0/252 (0.00)	4/894 (0.45)	187/3769 (4.96) ***
♀ “Fukushima-funen 1” ♂ “Oi 7	355/936 (37.93)	148/775 (19.10)	19/192 (9.90)	1/204 (0.49)	33/477 (6.92)	556/2584 (21.52) ***
♀ “Fukushima-funen 1” ♂ “S3-118(2)”	416/1017 (40.90)	258/987 (26.14)	42/293 (14.33)	2/152 (1.32)	71/884 (8.03)	789/3333 (23.67) ***
Total	1136/3716 (30.57) ***	874/4110 (21.27) ***	153/1555 (9.84) ***	3/656 (0.46) ***	109/2303 (4.73) ***	2275/12,340 (18.44)

**^$^**: All female and male parents had *ms1/ms1* and *Ms1/ms1* genotype, respectively; *** significantly different at *p* < 0.001 by post hoc analysis of Pearson’s Chi-squared test.

**Table 5 plants-10-00398-t005:** Somatic embryogenesis (SE) initiation frequency from polycross-pollinated-derived seeds of sugi. The data represent the explants with SE initiation response and the total number of explants tested, and the number in the parentheses represent the initiation frequency (%) for each polycross family at different seed collection times.

Polycross Family ^1^	SE Initiation Frequency (%) by Seed Collection Time			
	Early-July	Mid-July	Late-July	Total
“S 11” × “3 Mix”	157/404 (38.86)	583/819 (71.18)	471/841 (56.00)	1211/2064 (58.67) ***
“S 11” × “10 Mix”	215/514 (41.83)	451/726 (62.12)	345/773 (44.63)	1011/2013 (50.22) ***
Total	372/918 (40.52) ***	1034/1545 (66.93) ***	816/1614 (50.56) ***	2222/4077 (54.50)

^1^: Mother tree and pollen parents used for polycross seed families are shown in Appendix A; *** significantly different at *p* < 0.001 by post hoc analysis of Pearson’s Chi-squared test.

**Table 6 plants-10-00398-t006:** Somatic embryogenesis (SE) initiation frequency of megagametophyte explants derived from seventeen male-fertile, eight male-sterile, and five polycross seed families of sugi. The data represent the explants with SE initiation response and the total number of explants tested, and the numbers in the parentheses represent the initiation frequency (%) for each seed family.

Seed Family ^1^	SE Initiation Frequency (%) of Seed Explants Derived From		
	Male-Fertile Family	Male-Sterile Family	Polycross Family
Chiyoda 327 (OP)	12/100 (12.00)		
Kuji 6 (OP)	9/60 (15.00)		
Kuji 9 (OP)	1/74 (1.35)		
Kuji 14 (OP)	545/1224 (44.53)		
Kuji 17 (OP)	17/264 (6.44)		
Kuji 39 (OP)	5/94 (5.32)		
Naka 3 (OP)	41/192 (21.35)		
Naka 5 (OP)	10/122 (8.20)		
Naka 6 (OP)	18/78 (23.08)		
Nihari 2 (OP)	71/426 (16.67)		
Taga 2 (OP)	191/502 (38.05)		
Taga 4 (OP)	37/192 (19.27)		
Taga 10 (OP)	2/24 (8.33)		
Taga 14 (OP)	195/580 (33.62)		
Tsukuba 2 (OP)	27/192 (14.06)		
Yamazaki 5 (OP)	23/156 (14.74)		
Yanase 104 (OP)	105/750 (14.00)		
“Shindai 3” × “Suzu 2”		496/1943 (25.53)	
“Fukushima-funen 1” × “S3-37(1)”		81/1056 (7.67)	
“Fukushima-funen 1” × “Oi 7”		355/936 (37.93)	
“Fukushima-funen 1” × “S3-118(2)”		416/1017 (40.90)	
“S1S1-35” × “Gosenshi 1”		10/74 (13.51)	
“S1S1-23(1)” × “Gosenshi 1”		1/36 (2.78)	
“S1S1-10(1)” × “Gosenshi 1”		54/186 (29.03)	
“S1S1-51(1)” × “Gosenshi 1”		14/347 (4.03)	
“S 1” × “3 Mix”			32/286 (11.19)
“S 1” × “9 Mix”			28/268 (10.45)
“S 11” × “3 Mix”			1301/2280 (57.06)
“S 11” × “10 Mix”			1031/2068 (49.85)
“G 1” × “10 Mix”			40/478 (8.37%)
Total	1309/5030 (26.02) ***	1427/5595 (25.50) ***	2432/5380 (45.20) ***

^1^: Mother trees and pollen parents used for polycross seed families are shown in Appendix A; OP: open-pollinated; *** significantly different at *p* < 0.001 by post hoc analysis of Pearson’s Chi-squared test.

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
