# Peer review of "Somatic Embryogenesis Initiation in Sugi (Japanese Cedar, Cryptomeria japonica D. Don): Responses from Male-Fertile, Male-Sterile, and Polycross-Pollinated-Derived Seed Explants"

_plants, 2021, doi:10.3390/plants10020398_

Round 1

Reviewer 1 Report

Maruyama et al. has presented important results on study of somatic embryogenesis in sugi. Important to note that several factors were evaluated. The paper does not really present novel concepts. However, obtained results are significant. The number and quality of references is appropriate and gives proper credit to related work. The overall presentation is well structured and clear generally. 

Author Response

Response to Reviewer 1

(C) Maruyama et al. has presented important results on study of somatic embryogenesis in sugi. Important to note that several factors were evaluated. The paper does not really present novel concepts. However, obtained results are significant. The number and quality of references is appropriate and gives proper credit to related work. The overall presentation is well structured and clear generally. 

(R) We would like to thank you for your valuable comments on our document. We are improving the content of the manuscript by modifying the points indicated by the other three reviewers. We hope that the results of our paper will also serve as a reference for other conifers.

Thanks again for your kind comments.

Best regards,

Tsuyoshi E. Maruyama

Reviewer 2 Report

As can be read in my comments I do not agree on the evaluation and presentation of the data. More valuable input can be obtained.

Remember italics for all latin names as in vitro

Table 1 – I am not sure it makes any sense to add the two treatments to a total number. The mid June result should be compared to the other timepoints on its own and there is a difference.

Table lines not in line with text and makes it difficult to read

Line 118; what did the other authors do differently?

Line 126; it should be more clear this is total figures, which I am not sure makes sense.

Line 127; this has to be specified to one of the species used.

Table 3; Average should be within a species not between all species. More value is added if evaluated on species level.

Table 4; Again, average of all species is not useful. There is a great gap in the results within species and average will not give the correct conclusion for all species. And a comparison and conclusion on species would be of value. Table and legend should be in the same page. As with the other tables difficult to read.

Table 5; same comment

Line 219; The cultures can be kept for years which is valuable but it is not clear whether plant can be developed as well from these cultures and whether this rate changes.

Line 232; a list of all species will be valuable here.

Table 6: It will help the reader if on one page only.

Author Response

Response to Reviewer 2

(C) As can be read in my comments I do not agree on the evaluation and presentation of the data. More valuable input can be obtained.

(R) We would like to thank you for your constructive comments. We have changed the manuscript accordingly as described in each point below. Modifications in the manuscript have been highlighted in red.

(C) Remember italics for all latin names as in vitro

(R) Checking and corrections were made throughout the manuscript.

(C) Table 1 – I am not sure it makes any sense to add the two treatments to a total number. The mid June result should be compared to the other timepoints on its own and there is a difference.

(R) The results of statistical analysis for the seed collection time (L93-96) and for culture media (L107-109) were added in the manuscript.

(C) Table lines not in line with text and makes it difficult to read

(R) The Tables lines were aligned with the text.

(C) Line 118; what did the other authors do differently?

(R) The paragraph (L125-130) was corrected as follows: Even though, different initial culture medium and seed source was used, the highest SE initiation frequencies in sugi (20%−to 35%) with seeds collected in mid-July was also reported by Ogita et al. [28]. Similarly, Taniguchi and Kondo [29] also reported that seeds collected in mid-July were the best explants for SE initiation in sugi, recording induction rates of up to 33.3%, testing 20 different families from the open-polinated (OP) seed orchard.

(C) Line 126; it should be more clear this is total figures, which I am not sure makes sense.

(R) We are specifying that the initiation frequencies are the means of all mother trees (L133-134).

(C) Line 127; this has to be specified to one of the species used.

(R) We are specifying that the initiation frequencies are the means of all mother trees (L133-134).

(C) Table 3; Average should be within a species not between all species. More value is added if evaluated on species level.

(R) As reviewer recommend, the average and the result of statistical analysis among seed families were included in Table 3 and in the manuscript (L167).

(C) Table 4; Again, average of all species is not useful. There is a great gap in the results within species and average will not give the correct conclusion for all species. And a comparison and conclusion on species would be of value. Table and legend should be in the same page. As with the other tables difficult to read.

(R) As reviewer recommend, the average and the result of statistical analysis among seed families were included in Table 4 and in the manuscript (L189). Table and legend will be on the same page.

(C) Table 5; same comment

(R) The result about the differences beetwen polycross seed families and seed collection times are included in Table 5 and in the manuscript (L227-229). Table and legend will be on the same page.

(C) Line 219; The cultures can be kept for years which is valuable but it is not clear whether plant can be developed as well from these cultures and whether this rate changes.

(R) We mention in the manuscript (L286-288) that although embryogenic lines can be subcultured for a long time, differences among lines were observed regarding their plant conversion capacity (data not shown). The discussion about plant conversion capacity will not be presented here due to it will be discussed in another report. We ask for your kind understanding.

(C) Line 232; a list of all species will be valuable here.

(R) We are mentioned in each section that seed families used are presented in their respective Tables. For example, for Male-Fertile-derived Seed Explants (Supplementary Table S1, Section 4.1.1), for Male-Sterile-derived Seed Explants (Supplementary Table S2, Section 4.1.2), and for Polycross-pollinated-derived Seed Explants (Supplementary Table S3, Section 4.1.3). In addition, we are adding the number of families in Table explanation.

(C) Table 6: It will help the reader if on one page only.

(R) Table and legend will be on the same page.

Again, thanks for your constructive comments regarding to improve the quality of the report.

Kind regards,

Tsuyoshi E. Maruyama

Reviewer 3 Report

The ms focuses on SE initiation in sugi and describes the results form several years of studies with various explant materials and many families. Although the scientific novelty is not high, these results are of practical use for SE propagation of sugi. The presentation and structure of the ms, however, need major improvement before it can be considered for publication. Furthermore, even though the focus is on SE initiation, inclusion of some more data now presented either as pending (fathers' contribution in polycross) or not shown (embryo production capacity of the resulting SE lines) would add a lot to the ms and improve its value remarkably, and is thus suggested.

More detailed comments:

Material & methods

  • Explant collection times: Did you record day degrees (d.d. sum) at the collection times ? This would be very helpful when comparing years and in determination of optimal collection time !
  • Supplementary tables were not achievable for me, so it was quite difficult to follow the experimental plans when only referring to those - I suggest that you add some Basic information, such as the number of families used int he different experiments into the text too
  • R249 -PGRs not told, are they same as described in 4.3 ?
  • R258 - describe the different explant types and their cold storage!! This part of the work is not told anywhere, except discussed in conclusions..
  • R273 - what was the explant type used in all the other experiments except in the one testing varying types ? Please add.  

Results & discussion versus conclusions

Please revise: First give the results, then discuss them and finally give shorter conclusions. Now it is mixed, and conclusions contain discussion (e.g. r 307-314) while the results and discussion section has very little actual discussion on some topics.

  • 2.1.1. was tested only with one family. There is variation e.g. in optimal timing among the families as also shown in the other experiments of this very same study - please note and discuss at the approriate point.
  • R101 - did the SE lines initiated with and wihtout PGRs proliferate equally ? Were they all subcultured with PGRs ?
  • Table 1... You did not continue collection sto see if even better response could be achieved later on ? But you used later collection times in other experiments ?
  • Fig 1: Is A-B on the medium with PGR ? Please clarify. Add information on how many weeks after initiation the photos were taken.
  • Table 2: Show explants types in title row (instead of 1,2, 3), not in the foot note
  • Fig.2. Add information on how many weeks after initiation the photos were taken.
  • Table 3: Add the number of families in the explanation.
  • Table 4. Show explant types and their cold storage in the title row instead of 1-5
  • R204-208 Please add these results, that would be very interesting and improve the value of the ms
  • R226 please shortly give these results. Were there differences beteeen the lines derived from male sterile or male fertile explants ? Or did you observe differences in the enbryo production capacity also among the SE lines having different morphology ?

Author Response

Response to Reviewer 3

(C) The ms focuses on SE initiation in sugi and describes the results form several years of studies with various explant materials and many families. Although the scientific novelty is not high, these results are of practical use for SE propagation of sugi. The presentation and structure of the ms, however, need major improvement before it can be considered for publication. Furthermore, even though the focus is on SE initiation, inclusion of some more data now presented either as pending (fathers' contribution in polycross) or not shown (embryo production capacity of the resulting SE lines) would add a lot to the ms and improve its value remarkably, and is thus suggested.

(R) We would like to thank you for your valuable comments regarding to improve our manuscript. We have changed the manuscript accordingly as described in each point below. Modifications in the manuscript have been highlighted in red.

More detailed comments:

Material & methods

(C) Explant collection times: Did you record day degrees (d.d. sum) at the collection times? This would be very helpful when comparing years and in determination of optimal collection time!

(R) We absolutely agree with you about that the information of record day degrees would be very useful to compare seed collection years and determine the optimal collection time. However, unfortunately we did not record this information. We will consider this recommendation for our next experiments.

(C) Supplementary tables were not achievable for me, so it was quite difficult to follow the experimental plans when only referring to those - I suggest that you add some Basic information, such as the number of families used int he different experiments into the text too

(R) As reviewer recommend, we are adding information in the Table explanation and also in the text (Materials and Methods) about the number of seed families used in each experiment.

(C) R249 -PGRs not told, are they same as described in 4.3 ?

(R) Although we mentioned that for further SE initiation experiments, medium supplemented with 10 μM 2,4-dichlorophenoxyacetic acid (2,4-D) and 5 μM 6-benzylaminopurine (BA) were used (L109-110), we are adding this mention also in the Materials and Methods section (L345-346).

(C) R258 - describe the different explant types and their cold storage!! This part of the work is not told anywhere, except discussed in conclusions.

(R) The different initial explant types were added in the Materials and Method section also (L320-323), and related discussion from the Conclusions section were move to the Results and Discussion section (L202-213).

(C) R273 - what was the explant type used in all the other experiments except in the one testing varying types? Please add.  

(R) We are specifying in the section 4.1. Plant Material, that except to experiments with different initial explant types, in all the other experiments the entire megagametophyte was used as initial explant (L302-303).

Results & discussion versus conclusions

(C) Please revise: First give the results, then discuss them and finally give shorter conclusions. Now it is mixed, and conclusions contain discussion (e.g. r 307-314) while the results and discussion section has very little actual discussion on some topics.

(R) As reviewer recommend, the discussion of the results was expanded including also the results among seed families as recommended for another reviewer. Part of the discussions including Table 6 from the Conclusions section were move to the Results and Discussion section, and shorter conclusive points including the proposal of the directions for the future studies were added in to Conclusion section.

(C) 2.1.1. was tested only with one family. There is variation e.g. in optimal timing among the families as also shown in the other experiments of this very same study - please note and discuss at the approriate point.

(R) This observation is correct. This was our first experiment in Sugi, which was the beginning of a series of subsequent experiments shown in our report. As a first experiment, the results with only one seed family indicated that there were differences in the time of collection of the seeds. These results were later confirmed with other male fertile, male sterile, and polycross derived seed families, as presented in our report. However, although differences regarding to SE initiation frequencies among families were observed throughout our experiments, the optimal collection time for almost all families was determined around mid-July.

(C) R101 - did the SE lines initiated with and wihtout PGRs proliferate equally? Were they all subcultured with PGRs?

(R) ECLs initiated from medium with or without PGRs were subsequently subcultured on medium with PGRs. They proliferated similarly regardless of origin.

(C) Table 1... You did not continue collection sto see if even better response could be achieved later on? But you used later collection times in other experiments?

(R) This observation is correct. As our first experiment in Sugi, we set the collection time from mid-June to mid-July. In subsequent experiments we set the collection time from the early-July to late-July.

(C) Fig 1: Is A-B on the medium with PGR? Please clarify. Add information on how many weeks after initiation the photos were taken.

(R) As reviewer recommend, we are adding information about the medium and culture time in Fig.1, as follows: Figure 1. Somatic embryogenesis (SE) initiation in sugi, (A) extrusion of embryogenic cells (ECs) about 2 weeks after culture initiation on medium with plant growth regulators (PGRs), (B) ECs proliferation on medium with PGRs approximately 4 weeks after extrusion, and (C) germination and SE initiation on medium with no PGRs approximately 4 weeks after culture initiation. Bars: 5 mm.

(C) Table 2: Show explants types in title row (instead of 1,2, 3), not in the foot note.

(R) As reviewer recommend, we are show explants types in title row (instead of 1,2, 3), not in the foot note.

(C) Fig.2. Add information on how many weeks after initiation the photos were taken.

(R) As reviewer recommend, we are adding information about the medium and culture time in Fig.1, as follows: Figure 2. Somatic embryogenesis (SE) initiation in sugi, (A) embryogenic cells (ECs) proliferation from whole seed explant on medium with plant growth regulators (PGRs) approximately 6 weeks after culture initiation and, (B) from seed with coat cut lengthwise on medium with PGRs approximately 4 weeks after culture initiation. Bars: 1 mm.

(C) Table 3: Add the number of families in the explanation.

(R) As reviewer recommend, we are adding the number of families in Table explanation.

(C) Table 4. Show explant types and their cold storage in the title row instead of 1-5.

(R) As reviewer recommend, we are show explants types in title row (instead of 1-5), not in the foot note.

(C) R204-208 Please add these results, that would be very interesting and improve the value of the ms.

(R) Sorry, the results of this study are in preparation to be published soon. To respect copyright, we are unable to present these results prior to publication. We ask for your kind understanding.

(C) R226 please shortly give these results. Were there differences beteeen the lines derived from male sterile or male fertile explants? Or did you observe differences in the enbryo production capacity also among the SE lines having different morphology?

(R) We mention in the manuscript (L286-288) that although embryogenic lines can be subcultured for a long time, differences among lines were observed regarding their plant conversion capacity (data not shown). In the same way, although the differences between male sterile and male fertile explants, and also between ECLs having different morphology can be observed, for our understanding and mentioned in our conclusions, these differences can be attributed to the genotype of plant material regardless of origin. Anyway, the discussion about plant conversion capacity will not be presented here due to it will be discussed in another report. We ask for your kind understanding.

Thanks again for your valuable comments in order to improve the quality of the manuscript.

Best regards,

Tsuyoshi E. Maruyama

Reviewer 4 Report

Dear Editors,

Thank you so much for choosing me as a reviewer of the manuscript entitled “Somatic embryogenesis initiation in sugi (Japanese Cedar, Cryptomeria japonica D. Don): responses from male-fertile, male-sterile, and polycross-pollinated-derived seed explants”. I hope that my comments will help Authors to improve their manuscript.

Detailed remarks concerning manuscript.

Abstract.

Should be improved. The background of the studies as well as the clear scientific hypothesis of the studies and purpose of the report should be given. The short information concerning methodology and concessive conclusions should be given.

SE initiation frequencies varied from 1.35% to 58.67%.” Please do not start the sentence with the abbreviation.

Key words. It is not recommended to use as key words the same words or phrases that appeared in the title of the manuscript.

All tables and figures should be clear for the reader without the referring to the text of the manuscript and they should contain all needed explanations.

Conclusions

It should be improved. They are too long and should not contain tables and reference citation. It should be a brief statement based on the obtained (presented) results. It should also give a clear answer for the question asked as scientific hypothesis. In conclusions may also contain the proposal of the directions for the future studies. Changes in the conclusions imply changes in the section “Results and discussion’’. This changes concerning the description and discussion of the results (tables) presented in the conclusions. Discussion should be a little bit expanded. Moreover, I suggest to Authors include in discussion their own opinion concerning interpretation of obtained results.

Reference list.

Please go through the whole reference list and improve the editorial mistakes. For example. Some Journal title are abbreviated but some not. See Reference no 33 Cairney, J.; Pullman, G.S. The cellular and molecular biology of conifer embryogenesis. New Phytologist 2007, 176, 511–536.

Author Response

Response to Reviewer 4

(C) Dear Editors,

Thank you so much for choosing me as a reviewer of the manuscript entitled “Somatic embryogenesis initiation in sugi (Japanese Cedar, Cryptomeria japonica D. Don): responses from male-fertile, male-sterile, and polycross-pollinated-derived seed explants”. I hope that my comments will help Authors to improve their manuscript.

(R) We would like to thank you for your valuable comments regarding to improve our manuscript. We have changed the manuscript accordingly as described in each point below. Modifications in the manuscript have been highlighted in red.

Detailed remarks concerning manuscript.

Abstract.

(C) Should be improved. The background of the studies as well as the clear scientific hypothesis of the studies and purpose of the report should be given. The short information concerning methodology and concessive conclusions should be given.

(R) As reviewer recommend, we are changing the Abstract including the purpose of the report, short information concerning methodology and conclusion.

(C) “SE initiation frequencies varied from 1.35% to 58.67%.” Please do not start the sentence with the abbreviation.

 (R) We correct as follows: Initiation frequencies depending of the plant genotype varied from 1.35% to 57.06%.”

(C) Key words. It is not recommended to use as key words the same words or phrases that appeared in the title of the manuscript.

 (R) As reviewer recommend, we are changing the key words with phrases that not appeared in the title of the manuscript.

(C) All tables and figures should be clear for the reader without the referring to the text of the manuscript and they should contain all needed explanations.

(R) As reviewer recommend, we are improved the explanation of Tables and Figures, including information of number of seed families, type of explants, culture media, and culture time, and also the results among seed families as recommended for another reviewer.

(C) Conclusions

It should be improved. They are too long and should not contain tables and reference citation. It should be a brief statement based on the obtained (presented) results. It should also give a clear answer for the question asked as scientific hypothesis. In conclusions may also contain the proposal of the directions for the future studies. Changes in the conclusions imply changes in the section “Results and discussion’’. This changes concerning the description and discussion of the results (tables) presented in the conclusions. Discussion should be a little bit expanded. Moreover, I suggest to Authors include in discussion their own opinion concerning interpretation of obtained results.

 (R) As reviewer recommend, the discussion of the results was little bit expanded. Part of the discussions including Table 6 and references from the Conclusions section were move to the Results and Discussion section, and shorter conclusive points including the proposal of the directions for the future studies were added in to Conclusion section.

(C) Reference list.

Please go through the whole reference list and improve the editorial mistakes. For example. Some Journal title are abbreviated but some not. See Reference no 33 Cairney, J.; Pullman, G.S. The cellular and molecular biology of conifer embryogenesis. New Phytologist 2007, 176, 511–536.

(R) We checked and corrected mistakes and the order of the references after revision of the manuscript.

Thanks again for your valuable comments in order to improve the quality of the manuscript.

Kind regards,

Tsuyoshi E. Maruyama

Round 2

Reviewer 2 Report

Thank you for taking my comments into consideration. 

Author Response

Dear Reviewer,

Thank you for all your valuable comments to improve our manuscript.

Best regards

Tsuyoshi E. Maruyama

Reviewer 3 Report

The ms has improved from its first version, and the authors have answered to almost all of my comments.

My original suggestion was to include molecular data covering the participation of different father trees into the progeny of polycross mixes, potentially explaining also the SE initiation success (Line 239) , as well as data covering varying embryogenic potential / plant conversion ability of the sugi genotypes (L287). As the authors answered that these results are currently being published in other papers, I kindly ask them to add references to these papers / manuscripts, since these are some of the most interesting points in this ms.  With this addition, I see that the ms can be accepted for publication. 

Author Response

Reviewer # 3 (Second Round)

Dear Reviewer,

As reviewer recommended, we are adding the respective references regarding Line 239 and Line 287, and corrected the order of the references after addition. Modifications in the manuscript have been highlighted in red.

Thank you for all your valuable comments to improve our manuscript.

Best regards

Tsuyoshi E. Maruyama